# Detection and Quantification of Bisphenol A in Surface Water Using Absorbance–Transmittance and Fluorescence Excitation–Emission Matrices (A-TEEM) Coupled with Multiway Techniques

**DOI:** 10.3390/molecules28207048

**Published:** 2023-10-12

**Authors:** Thomas Ingwani, Nhamo Chaukura, Bhekie B. Mamba, Thabo T. I. Nkambule, Adam M. Gilmore

**Affiliations:** 1Institute for Nanotechnology and Water Sustainability, College of Engineering, Science and Technology, University of South Africa, Johannesburg 1709, South Africa; 65102924@mylife.unisa.ac.za (T.I.); mambabb@unisa.ac.za (B.B.M.); adam.gilmore@horiba.com (A.M.G.); 2Department of Physical and Earth Sciences, Sol Plaatje University, Kimberley 8300, South Africa; nchaukura@gmail.com; 3Horiba Instruments Incorporated Inc., Piscataway, NJ 08854, USA

**Keywords:** method development, optimisation, and validation, parallel factor modelling, partial least squares modelling

## Abstract

In the present protocol, we determined the presence and concentrations of bisphenol A (BPA) spiked in surface water samples using EEM fluorescence spectroscopy in conjunction with modelling using partial least squares (PLS) and parallel factor (PARAFAC). PARAFAC modelling of the EEM fluorescence data obtained from surface water samples contaminated with BPA unraveled four fluorophores including BPA. The best outcomes were obtained for BPA concentration (R^2^ = 0.996; standard deviation to prediction error’s root mean square ratio (RPD) = 3.41; and a Pearson’s r value of 0.998). With these values of R^2^ and Pearson’s r, the PLS model showed a strong correlation between the predicted and measured BPA concentrations. The detection and quantification limits of the method were 3.512 and 11.708 micro molar (µM), respectively. In conclusion, BPA can be precisely detected and its concentration in surface water predicted using the PARAFAC and PLS models developed in this study and fluorescence EEM data collected from BPA-contaminated water. It is necessary to spatially relate surface water contamination data with other datasets in order to connect drinking water quality issues with health, environmental restoration, and environmental justice concerns.

## 1. Introduction

Bisphenol A (BPA) (Table 1), which belongs to the alkyphenol homologous series, is used in the production of polycarbonate plastics and epoxy resins [1,2], which are used to make hard plastic items like storage containers [1] and as linings to cover the interiors of metal products [2]. The aquatic, terrestrial, and atmospheric environments are all impacted by BPA due to its widespread use [2]. Using consumer products that contain BPA can also contaminate the environment and food. Given that BPA is a compound that disrupts the endocrine system and is toxic to the reproductive, developmental, and systemic systems [3], its levels in aquatic systems need to be tracked and monitored. Hence, there is a need for a cost-effective and simple-to-use technique to characterise BPA contamination in water.

Some previous studies have used FEEM in conjunction with a number of multiway techniques to detect and quantify BPA in different plastic materials [4,5,6]. In one study, BPA and 4-nonylphenol in samples of different plastic materials that were in contact with beverages and/or food were identified using fluorescence EEM in conjunction with kinetic third-order or four-way PARAFAC data modelling. Four-way PARAFAC modelling provided satisfactory outcomes for BPA analysis. Third-order methods have the advantages of being able to quantify analytes in the presence of uncalibrated compounds and solve systems with very poor selectivity [7,8]. The current study combined EEM data from surface water laced with standard solutions of BPA with three PARAFAC analysis modes that were restricted to non-negative values in order to identify BPA and other fluorescent components in surface water laced with BPA. In order to calculate BPA, we used Origin 8.6 to regress the data imported from PLS modelling. The A-TEEM method has the advantage that a simultaneous acquisition of absorbance and fluorescence spectral data in a single instrument makes it easier to correct fluorescence IFEs and produce fluorescence spectral data free from IFEs [9].

While another study [5] proposed a fluorimetric method in conjunction with the second order calibration of EEMs for the evaluation of the BPA migration test, the current study employed the PARAFAC and PLS models, respectively, to detect and quantify BPA in surface water. The method used in the previous study was capable of identifying and quantifying BPA with a high degree of certainty because of the trilinearity of the data tensor, which made sure that the solution obtained through PARAFAC analysis was distinct. Consequently, even though other fluorophores were present in the test sample, one element of the decomposition matched up with BPA. The advantage of the PARAFAC model in the current study, however, is that instrument selectivity is not required when using multiway data [10]. The study used the Q residual versus Hotelling T^2^ plot only, whereas the current study used the predicted and measured BPA concentration, scores on LV1 versus scores on LV2, and leverage versus studentized residuals plots in addition to the Q residual versus Hotelling T^2^ plot for the identification of outlier samples during PARAFAC calibration. This suggests that the current study found more outliers overall and a greater variety of them than the earlier study. In comparison to the second prior study [5], PARAFAC models were validated in the current study more thoroughly. This is due to the fact that the previous study used the core consistency diagnostic only to establish the ideal number of components in the developed model, whereas in the present study, the split-half analysis, explained variance, and visualisation of spectral loadings were evaluated in addition to the core consistency diagnostic technique to validate the PARAFAC models. In contrast to the prior study [5], which quantified BPA using PARAFAC calibration, the current study used the PLS algorithm in conjunction with Origin 8.6 software to ascertain the concentration of BPA in surface water samples.

A related study [6] combined FEEM and second-order chemometrics methods based on the alternating trilinear decomposition algorithm to ascertain the presence of BPA in polycarbonate plastics. Second-order calibration has the benefit of allowing analytes to be quantitated even when there are unmodelled interferents. Instrument selectivity is not required when using multiway data, which is a benefit of using PARAFAC modelling [10]. As a result, different algorithms were employed by the methods to interpret EEM data.

The fluorescence excitation-emission matrices (FEEM) method, which has a short run time [11], is straightforward to use, is cost-effective [12], and does not require the use of extraction or reagents, includes the simultaneous absorbance and excitation–emission matrices (A-TEEM) in combination with multivariate analysis techniques. Based on a review of the literature and the distribution of species sensitivities, it was found that the measured concentration of BPA in surface water ranged from 100 to 400 µM in Europe and North America [13]. In Asia, an upper limit of 0.014 g/L for the concentration of BPA in surface water was recorded [14]. In South African rivers, the concentration of BPA ranged from non-detectable to 89 ng/L [15]. Based on a review of the literature and the distribution of species sensitivities, it was found that the measured concentration of BPA in surface water was 273 ± 9.73 ppb [16]. The median concentration in various surface water samples from Latin American nations was 0.53 ng/mL, while the min–max concentration ranged from 0.09 to 1.46 ng/mL [17].

Due to the lack of specificity in fluorescence spectroscopy, the trilinear parallel factor (PARAFAC) model was used to detect the presence of BPA, and the partial least squares (PLS) model was used to predict the concentration of BPA in spike surface water validation samples. Since the convolution of different component fingerprints in a multicomponent mix containing environmental samples results in complicated, difficult-to-interpret EEM spectra, fluorescence spectrometers and multivariate analyses are commonly used together [18].

The PARAFAC model can be used to interpret fluorescence EEMs at various excitation and emission wavelengths [19]. When the correct number of components for a PARAFAC model is specified, PARAFAC can resolve the correct emission and excitation spectra of each constituent in a mixture. It is necessary that the excitation and emission spectra be independent and linear, and that the concentrations of the analytes vary independently [18]. The component concentration scores, emission loadings, and excitation loadings obtained during PARAFAC modelling were used for the classification of components in the surface water samples spiked with BPA.

The PLS model can be used to apply linear regression in order to compare the measured and predicted analyte concentrations (X-block and Y-block data, respectively), as well as to classify EEM data [20,21].

Due to the difficulty of conducting laboratory experiments on surface water that has not been spiked and the highly variable composition of surface water as well as the fact that alkylphenols are known to be unstable in water [22], the analyte is added to surface water samples during an analysis. In order to account for matrix effects that have an impact on the analytical response, matrix matching is used in the analysis. Furthermore, the accuracy of tagging components with fingerprints is increased by analysing both the CDOM components and the BPA standards in the same solutions.

Even though fluorescence spectroscopy coupled with PARAFAC and PLS algorithms can be used to analyse BPA in water, it is still challenging to quantify BPA due to its varying fluorescence intensities and complex underlying interference in water. For instance, if BPA is found in aqueous solutions at concentrations of more than 10 µM, there may be cause for concern due to the interference that lies beneath [23]. BPA has a much lower fluorescence quantum yield of 0.002 [23] in comparison to many DOM components with fluorescence quantum yields ranging between 0.008 and 0.016 [24]. Thus, BPA is less fluorescently active than many DOM components. Many of these DOM components have surface water DOM concentrations of 1 to 20 ppm [11].

In the present study, for the quick (run time: less than 4 min) extraction- and reagent-free detection and quantitation of BPA in common surface water, we tested a possibly sensitive method at lower µM concentration levels. The technique should be used to measure the BPA levels in water due to how simple and inexpensive it is to use.

## 2. Results

The A-TEEM-PARAFAC-PLS analytical method had a short run time (<4 min), high sensitivity at lower µM levels for BPA, required little to no sample preparation, and was easy to use. This makes the method effective at monitoring BPA contamination in water.

### 2.1. Absorbance, Excitation, and Emission Spectra

The absorbance spectra showed three peaks at 206, 215, and 275 nm (Figure 1a), indicating the presence of BPA introduced into the sample. As a result of the addition of BPA, the excitation spectra of the surface water sample spiked with a BPA standard solution revealed two peaks at excitation wavelengths of 203 and 218 nm (Figure 1b), which are consistent with the excitation wavelength of 203 nm at which BPA was detected in a previous study [25]. Figure 1c demonstrates that the BPA emission peak happens at a wavelength of 306 nm. The maximum BPA emission wavelength is in agreement with that of an earlier study [25].

### 2.2. Excitation–Emission Matrix Signature for BPA

Figure 2a shows the fingerprint EEM of an unspiked (blank/baseline) surface water sample. The unspiked surface water sample showed fluorophores at 230–425 nm for the excitation wavelength and 325–600 nm for the emission wavelength, which indicated the presence of humic substances [26,27]. Figure 2b shows the fingerprint EEM of a surface water sample that had been spiked with BPA at a concentration of 21 µM. In the current study, BPA was found to have excitation and emission wavelengths ranging from 220 to 290 nm and 290 to 375 nm, respectively. These wavelengths match those at which BPA was detected in a previous study [28]. Low-concentration or low-quantum-yield fluorophores may not have visible peaks due to the dominating response of other fluorophores. This is especially true when the dominant wavelength regions coincide with the excitation and emission maxima of the minor ones. Therefore, it is possible that more than one fluorophore had an effect on the fluorescence spectra that were obtained here.

### 2.3. Construction and Validation of the PARAFAC Model

The fact that the captured variance obtained in this study using up to four components was higher than that obtained using more than four components shows that four components was the ideal number (Figure 3). An underfitted model typically has a low captured variance [29]. However, if there are too many components used, the model will be overfit. In this situation, the obtained model will capture nearly all of the variance [30]. The percent captured variance with three components can also be used with these types of data [31].

The core consistency for solutions based on a one-component model started out high (at 100%), but it abruptly decreased once the fifth component was fitted. The four components matched the number in the largest model that maintained a large core consistency [29]. The four-component model was the best fit for the dataset and generated the highest resolution of EEM spectra of components (Figure 3) for the 120 EEM dataset, with core consistency diagnostic scores for the one, two, three, four, and five components, respectively, of 100, 100, 100, 67, and <0. The core consistency value is almost 100% if the PARAFAC model is valid. Core consistency will be negative (or even very close to zero) if a trilinear model is unable to adequately describe the data or if there are too many components being fitted [32].

The results of the split-half analysis for the four-component model showed a similarity of 99.9% matching for both distinct splits, indicating that almost no distinction could be made between the constituents of each separate split set (Figure 4). At this stage, the results showed that the four-component PARAFAC model performed the best in terms of core consistency, percent explained variance, and similarity between split-half analyses.

The 2-D EEM contour plots for the four components in surface water samples spiked with BPA based on PARAFAC modelling of the EEM spectral data are shown in Figure 5. Component #1 was an organic material that resembled fulvic acid-like organic matter, with maximum Ex/Em wavelengths of ~260–325/440 nm (Figure 5a). Component #2, with maximum Ex/Em wavelengths of 265–375/500 nm, respectively, resembled humic acid-like organic matter (Figure 5b). Component #3 (Figure 5c), with maximum Ex/Em wavelengths of ~275/306 nm, respectively, is a representation of BPA. Component #4 resembled marine humic-like organic material and had maximum Ex/Em wavelengths of ~350/430 nm (Figure 5d).

Figure 5 also displays line plots for the four components of the excitation and emission spectral loadings determined via PARAFAC analysis. The correctly validated PARAFAC spectral loadings of excitation and emission represent measures of the pure analyte spectra when the fluorescence data are trilinear and the appropriate component numbers are applied [31]. These loadings were determined via PARAFAC based solely on the excitation and emission spectra of the fluorophoric components in the investigated solutions. Each fluorophoric component was represented by a single PARAFAC component, which was made up of PARAFAC scores for relative concentrations, an excitation loading related to the estimation of the pure excitation spectrum, and an emission loading related to the measurement of the plain emission spectrum. The model was reliable since the spectra were smooth and nearly identical [33]. The smooth, comparable, wide, and frequently unimodal emission loading spectra demonstrated the PARAFAC model’s viability [31]. On the other hand, it appears that the model was unsuccessful in identifying the excitation spectra of the pure chemical component for component #4, as indicated by the rather sharp excitation peak for that component at about 350 nm [31]. When the EEM data were analysed using the four approaches, the four-component model repeatedly reached the same conclusion.

The emission wavelengths for fulvic acid- and humic acid-like components matched those of components #1 and #2, respectively, which had previously been discovered to emit light at their maximum intensity at wavelengths of 440 and 500 nm [26,34,35]. The strongest emission was previously observed at 306 nm for BPA [25].

### 2.4. Construction of the PLS Model

The fluorescence spectra of 120 surface water samples containing BPA at concentrations ranging from 3 to 300 µM are shown in Figure 6a. On the basis of the visual interpretation of the peaks, the figure depicts one region with a maximum emission at about 306 nm. This maximum emission wavelength is typical of BPA and agrees with an earlier study [36]. Due to overlapping emissions from various substances present in the samples, self-absorption, energy transfer, and quenching phenomena, the spiked surface water samples displayed complex fluorescence properties as multicomponent solutions [37].

Three modes make up rPLS: namely, specified, suggested, and surveyed. The RMSECV values of 10.7, 10.8, and 11.8 µM for the specified, suggested, and surveyed modes, respectively, for the selected variables, show that the cross-validation errors were minimal. The specified mode was the default mode, which constructed the PLS model exclusively using the specified number of LVs and number of components. The suggested mode ran the PLS and cross-validation on the entire dataset, determining the most appropriate number of LVs, and then proceeded with the rPLS as in the specified mode. The PLS was run in the surveyed mode from 1 LV to the most possible LVs, and the set of results with the lowest RMSECV value was returned. Figure 6b shows how the RMSECV (blue curve) and RMSEC (green curve) parameters changed depending on how many LVs were used to create the prediction model. The residual errors for the samples used to train and validate the model were measured using the RMSEC, which, in the current study, had a typical value of 17.434 µM.

Figure 7a plots the predicted versus measured BPA concentration using 120 EEM spectra. The high value of the calibration coefficient of determination (R^2^ = 0.96) demonstrated the significant correlation between the predicted and measured BPA concentrations. The scores on LV1 versus the scores on LV2 plot (Figure 7b) revealed four outliers. According to the plot of Hotelling’s T^2^ statistic versus Q residuals reduced (Figure 7c), the scores were able to explain 99.35% of the total variance while the residuals still held onto 0.6% of it. The figure also demonstrates the absence of notable score outliers. Figure 7d shows that no significant cases existed [38].

The measured parameter values demonstrate that BPA in surface water can be analysed using the proposed method (Table 2). The RMSECV measures model performance in relation to validation samples, whereas the RMSEC measures model performance during the calibration (training) stage. The RMSEC and RMSECV values, which are 17.434 and 34.794 µM, respectively, are relatively low, indicating that the cross-validation and calibration errors are also low. Hence, the model is accurate. These variables, along with the percentage of variance explained, are indicators of how well a model performs in making predictions. The PLS model with five LVs was the best choice for the BPA regression analysis. The model explained 90.41% of the variance in the dependent variables and 98.84% of the variance in the predictors as a whole. Skewed estimates had no impact on the ability of the predictive model to make accurate predictions because of its low calibration, CV, and prediction biases [39]. The 120 EEM dataset had three outliers removed, and the PARAFAC model performed better as a result, according to the results. The values of R^2^_Cal_ and R^2^_CV_ both increased by 0.73% and 3.79%, respectively, which are small increments. This illustrated that the model was robust [40].

The plotted BPA concentration calibration curve and BPA validation curve are shown in Figure 8 as black dots and red diamonds, respectively. The performance indicators for the PLS modelling of the calibration and validation datasets are also shown in Figure 8. The high value of R^2^_Cal_ of 0.927, which denotes a strong correlation between the predicted and measured BPA concentrations, shows this.

The calibration and validation curves were found to be linear, and the high values of R^2^_Cal_, R^2^_CV_, and R^2^_Pred_—0.927, 0.677, and 0.832, respectively—demonstrated the robustness of the PLS model. Following the splitting of the 120 EEM data, the magnitudes of R^2^_Cal_ and R^2^_CV_ increased by 1.86% and 2.17%, respectively, demonstrating that the splitting of the 120-member dataset being modelled improved model performance and these results showed the robustness of the PLS model [41]. This improvement can also be attributed to the removal of outliers. Before deciding whether to include outliers in the regression analysis process or not, a very thorough analysis must be conducted. Because the RMSEP was so low (5.786 µM) and the R^2^_Pred_ was high (0.832), the predictive power of the model and the dependability were strong. The small biases obtained suggested that the A-TEEM-PLS method correctly measured and predicted the BPA concentration.

### 2.5. Validation of Spiked Samples

Figure 9a displays the predictions made by PLS modelling for the validation samples, i.e., surface water samples that were spiked with BPA at concentrations ranging from 50 to 270 µM. The PLS regression model generated a precise prediction model with constrained 95% prediction and 95% confidence bandwidths. Results from the model prediction revealed a strong correlation, with the R^2^ for the linear fit equalling 0.996. According to the R^2^ value, 99.6% of the fluorescence spectral data fit the PLS model. A random distribution of the points along the horizontal axis of the regular residuals versus the BPA concentration and consistent residuals of ±10 µM were observed in the residual plot (Figure 9b), indicating that the linear regression model correctly predicted the data [41].

The linear regression analysis report (Table 3) displays the linear regression parameters. The low residual sum of squares was evidence that the PLS model adequately fit the data. The low standard error of the intercept and slope values of 3.079 and 0.0167, respectively, that were obtained demonstrated the dependability of the analytical method and that the regression coefficients obtained were close to the actual coefficients. The R^2^, adj. R^2^, and Pearson’s r values of 0.996, 0.996, and 0.998, respectively, demonstrated a strong correlation between the predicted and measured BPA concentrations [42]. Considering that Pearson’s r value was positive, it was assumed that there was a positive correlation. The RMSE value for the prediction was 5.272 µM, while the mean absolute error (MAE) value was 4.378 µM, a difference of 0.894 µM (Table 3). Due to the fact that RMSE gives more weight to larger differences between values than MAE does, the dataset’s outliers can be blamed for the significant difference between MAE and RMSE (0.894 µM) [42].

The ANOVA table (Table 4) displays how the sum of squares and mean sum of squares are distributed based on the source of variation. The sum of squares and mean sum of squares have standard errors of 389.073 and 27.791 µM, respectively, showing that the independent and dependent variables measured and predicted by the regression model are not significantly different from one another. However, the sum of squares value illustrates that the model fits the data rather poorly. On the other hand, the analytical method was deemed significant due to its high F-value of 210.474 [43]. The *p*-value for the entire model test is Prob > F. The *p*-value is shown as Prob > F in the output of a regression model with m independent variables [44]. A *p*-value of zero, which was less than 0.05, indicated that the model and data had a significant goodness of fit. It further demonstrated the statistical significance of the analytical technique [45]. The null hypothesis, which states that none of the measured BPA concentrations were related to the predicted BPA concentrations, was rejected because the *p*-value was less than 0.05. In other words, variations in the predicted BPA concentration were consistent with variations in the measured BPA concentration.

### 2.6. Method Sensitivity and Limits of Detection and Quantification

The calculated detection and quantification limits for BPA in surface water were 3.512 and 11.708 µM, respectively. These were the lowest BPA concentrations that the A-TEEM-PLS analytical technique could accurately identify and quantitate. Since the lowest concentration of a measure that can be accurately measured by an analytical method is referred to as the lower limit of detection or sensitivity, then the method sensitivity is 3.512 µM. However, the method used in this study had a lower sensitivity than the method used in a prior study to determine BPA at low concentrations in naturally occurring mineral water kept in plastic bottles using HPLC with UV detection [46].

### 2.7. Recovery and Accuracy

The accuracy was measured as a percent recovery in accordance with the ASTM [37,41] and ICH [43] recommendations. An average method accuracy of 97.55% was obtained from the recovery determined by analysing three different BPA concentrations in surface water (at concentrations of 50, 180, and 270 µM) (Table 5). This suggests that the A-TEEM-PLS method is highly accurate because the results were, on average, 97.55% of what the theoretical calculation would have predicted.

## 3. Materials and Methods

### 3.1. Materials and Reagents

For use in this study, BPA with a 97% purity level, AR-grade methanol, and 0.45-micron GMF filters were provided by Sigma-Aldrich^®^, Modderfontein, Johannesburg, South Africa. Using an Elix Integral 10 water purification system (Merck Millipore, Germiston, East Rand, South Africa), deionised water was produced on-site.

### 3.2. Sampling

Using the grab sampling technique [47], surface water samples were taken from the Florida stream in Johannesburg, South Africa, during the winter of 2022 (26.1739° S, 27.8971° E). Clean 1 L amber glass bottles were used to collect the surface water samples. Following this, the samples were kept at 4 °C prior to being analysed for their absorbance and fluorescence properties within 48 h of sampling. The samples were then warmed to room temperature, filtered to remove larger bacteria and particles using 0.45-micron GMF filters, and spiked with BPA right before the acquisition of absorbance and fluorescence EEM spectra.

### 3.3. Preparation of a Stock Solution and an Intermediate Standard Solution

With a purity level of 97%, 11.77 g of BPA was weighed on a Mettler Toledo analytical balance (Greifensee, Switzerland) and then dissolved in 1 L of methanol to prepare a BPA stock solution with a concentration of 0.05 M. To prepare an intermediate standard solution of BPA at 1000 µM, 2 mL of the stock solution was diluted with AR-grade methanol in a 100 mL volumetric flask.

### 3.4. Sample Preparation

Prior to use, the sample cuvette for the Aqualog^®^ spectrometer (HORIBA Scientific Inc. Piscataway, NJ, USA) was thoroughly cleaned by submerging it completely for 12 h in 50% aqueous nitric acid and then rinsing it with deionised water to lessen background contributions. In order to prepare 100 calibration samples with BPA concentrations between 3 and 300 µM and 20 validation samples with BPA concentrations between 15 and 300 µM, the same quartz cuvette was filled with aliquots of surface water that had been filtered through a 0.45 µm GMF filter along with the BPA standards. The concentrations of the BPA standards in both sample types were different but evenly spaced. The total volume of the cuvette in its finished state was 4000 µL. After all the aliquots were placed inside the cuvette, the contents were vigorously shaken to ensure sample homogeneity. A green analytical chemistry protocol was applied to solve the problem of a weakly fluorescent system with significant component emission spectrum overlap. Because methanol has the potential to change the solution’s refractive index and contains a fluorescent background, it is called for, among other things, to limit the final methanol concentration in the samples to less than 2% in the cuvette [11].

### 3.5. Total Organic Carbon Determination

A Tekmar-Teledyne TOC analyser, which is based on UV-catalysed persulphate digestion to produce carbon dioxide, was used to measure the total organic carbon (TOC) in the surface water. Carbon dioxide was then detected using a nondispersive infrared detector.

### 3.6. The Calibration of the A-TEEM Instrument

The A-TEEM spectrometer autocalibrates, implying that the system initialises the drives of its monochromators, finds each drive’s home position, and assigns a wavelength value to this position using data from a calibration file. However, a sealed cuvette of Raman water was also scanned to check the calibration and throughput while also normalising EEM data.

### 3.7. Instrumentation and Software

The A-TEEM spectrometer (HORIBA Aqualog^®^ Yobin Yvon model UV-800C, Piscataway, NJ, USA) was used to collect the EEM spectral data of a surface water sample that had not been spiked and surface water samples that had been spiked with BPA standard solutions. The sample queue method was employed to collect EEM data. The acquired EEM spectra served as fluorescent fingerprints. The instrumental parameters used were a fixed 5 nm optical slit width, emission wavelengths from 245.21 to 827.32 nm spaced at 8 pixels (4.66 nm), and excitation wavelengths from 200 to 800 nm spaced at 5 nm.

The signal-to-noise ratio (S/N) has a big impact on how well data are acquired. A suitable integration time of 0.5 s and a medium CCD gain were applied to improve the S/N. To avoid CCD saturation during data collection, the signal was kept within the linear range of the detector. Quick and simultaneous EEM spectral data acquisition was made possible by the device, which used a thermoelectrically cooled CCD spectrograph detector to capture the entire emission spectrum at each excitation increment. A saturation mask width of 16 nm was used to lessen sensitivity to erroneous CCD signal saturation alerts and Rayleigh scatter width. Silicon photodiodes were used for the reference and absorbance detectors.

### 3.8. Multiway Data Analysis

Large datasets can be analysed using multiway data analysis techniques by encoding the data as a multidimensional array. The Eigenvector Solo software versions 8.7 and 8.6, on which the PARAFAC and PLS multiway analyses were based, were hyphenated to the A-TEEM spectrometer and were used to interpret all of the EEMs in the current study (Figure 10). The stand-alone algorithms were run on PLS_Toolbox.

#### 3.8.1. Optimisation of the PARAFAC and PLS Models

The experimental results were optimised due to various software and hardware conditions as well as light scattering and spectrum modifications that can result in artifacts. Optimising data acquisition improved spectral resolution. This was accomplished by using a large sample size, filtering the samples through 0.45-micron GMF filters, meticulously setting up the spectrofluorometer (Section 2.7), and preprocessing the EEM data acquired.

In order to interpret significant results, a large sample size (i.e., 100 calibration samples and 20 validation samples) was used to allow for a more accurate measurement of the treatment effect [48]. Therefore, a large sample size could optimise the PARAFAC and PLS models because the predictive power of the model increased as more components were added to it [49].

To more effectively classify the data, data preprocessing techniques like feature extraction, dimension reduction, scaling to the reference detector, spectral correction, dark signal subtraction, blank subtraction, and normalisation [50] were used, utilising the A-TEEM spectrometer software (V 3.6). To enhance the performance of the classifier, preprocessing aimed to identify the most informative set of features.

To maximise the number of fluorescent components in the samples analysed, the fits of the one- to five-component PARAFAC models of the EEM data were investigated. Each of the five models was evaluated for appropriateness using the percent captured variation, spectral loading visualisation, split-half analysis, and core consistency techniques as outlined in Section 3.8.2 [31,51].

The optimisation of variable selection and latent variables (LVs) was undertaken during PLS modelling as described earlier. The root mean square error of cross-validation (RMSECV) optimised the number of LVs. This was necessary because an insufficient number of LVs would indicate an insignificant relationship between the two variables.

Outliers were identified during PLS modelling by examining four types of sample/score plots: namely, the predicted and measured BPA concentration, scores on LV1 versus scores on LV2, Hotelling’s T^2^ statistic versus Q residuals reduced, and leverage versus studentized residuals plots. The outliers were tentatively removed as they can affect the overall quality of the model, and this is a crucial stage in the design of an optimised model [52]. These plots operate in the manner specified in Section 3.8.4.

#### 3.8.2. Construction of the PARAFAC Model

PARAFAC modelling was undertaken for the 120 EEM dataset. To rectify any systematically biased data; remove interference from inner-filter effects (IFEs), Raman scatter, and Rayleigh scatter; and normalise datasets with substantial intensity variations between samples, the EEM data were preprocessed. Following the importation of the EEM data from the A-TEEM instrument to the Solo software (V 9.0), the first- and second-order Rayleigh scatter, primary and secondary Raman scatter, and inner-filter effects were corrected using customised functions in the Aqualog^®^ spectrometer software (3.6). EEM filtering was performed by setting the first-order Rayleigh filter to 16 nm and the second-order Rayleigh filter to 32 nm. The filter half-width was set to 16 nm and the default Raman shift of 3382 cm^−1^ was used to mask the Raman scatter. The fluorescence intensities of the corrected EEM were normalised into Raman Units (R.U.) based on the peak area obtained using the same spectrometer to measure the Ex/Em 350 nm/396.5 nm 2-D spectrum of a sealed Raman water cuvette. To prevent fluorescence peaks from overlapping or approaching Raman or Rayleigh scattering, the bandwidth was varied [19].

Since the concentrations, absorptivities, and other physical characteristics of chemical compounds are mostly positive [19], non-negativity-constrained PARAFAC models were built using one to five components. Thus, all problem values were equal to or greater than zero. The use of the non-negativity constraint significantly reduced the feasible space of the parameters to be evaluated. The captured variance, core consistency, split-half evaluation, and an analysis of the spectral loading visualisation approaches were used to assess the validity of the PARAFAC models fitted.

#### 3.8.3. PARAFAC Model Validation

Split-half validation analysis was used to verify consistency within the dataset and was carried out by splitting the EEM dataset (in relation to samples) into two equal separate datasets and subjecting them to PARAFAC modelling. The two halves were modelled independently, and their outcomes were compared to determine their similarities as well as their similarities with the modelling results of the whole EEM dataset.

In this study, by observing that an increase obtained with more than the optimal number of components is modest compared to an increase in the explained variance obtained using up to the optimal number of components, the explained variance technique was used to determine the ideal number of components [31]. Underfitting will occur if there are too few components extracted; this can be quickly identified by evaluating the explained variance, since a model that is not well fit usually has a decreased explained variance. On the other hand, using an excessive number of components will overfit the model. The obtained model in this scenario will have nearly 100% explained variance.

The ideal number of components was determined to match the largest model’s number while maintaining a high level of core consistency [29]. A core consistency value close to 100 indicates a well-described model, while significantly lower values point to superfluous components. The fluorescence spectra of the samples were visually inspected for broad and frequently unimodal peaks, making the visual appearance of the loadings a helpful diagnostic [31]. The ideal number of components and, consequently, the suitability of the model had to be determined by striking a balance between split-half validation, explained variation, core consistency, and spectral loadings [30].

#### 3.8.4. Construction of the PLS Model

The spectral (X-block) calibration dataset was preprocessed for regression and classification via mean-centring and clutter removal using the full-rank extended mixture model. Concentration (Y-block) calibration data were also preprocessed via mean-centring. The emission spectra of 120 surface water samples spiked with BPA were plotted to show the maximum emission wavelength of BPA. In order to obtain accurate data analysis results by removing noise or unwanted signals, the preprocessing of the emission spectral data involved the exclusion of emission data just larger than 500 nm and higher [53].

The emission spectra and concentration values for the 120 samples from the X-block were copied and loaded into a new Y-block. The SIMPS algorithm from the Solo package (version 8.6) was used to calculate the PLS model. Each value derived from the calibration dataset and sample datasets was mean-centred then autoscaled. Preprocessing of the X- and Y-block data was carried out to extract useful data. The extended full-rank mixture model, including mean-centring and clutter eradication, was used to perform additional preprocessing on the dataset for spectral calibration (X-block). Additionally, mean-centring was used as a preprocessing step for the concentration calibration data (Y-block). The confusion matrix’s specifications created for the extraction requirements (*p* > 0.5) and the most likely classification method in accordance with the overall successful identifications were used to sort the data.

The model was made accurate and complete by performing variable selection to find every variable that affected the outcome. By eliminating extraneous variables that increased model complexity and decreased precision, a small number of variables were chosen using the recursive PLS (rPLS) algorithm. According to the regression coefficient calculated by the PLS model after each iteration, the variables were weighted recursively in rPLS. Initial spectral data were subjected to the variable selection procedure in order to determine the most useful variables and eliminate insignificant variables [54]. The selection process was carried out using the Solo package (version 8.6) from Eigenvector Inc.’s PLS_Toolbox. The cross-validation algorithm was used to iterate until the minimum RMSECV and maximum correlation coefficient were attained.

The RMSECV, which frequently has a minimum whose prominence varies with the noise in the data, was used to determine the ideal number of LVs [54]. As a result, this recursive validation was used to determine the number of LVs to use in the model.

The model was calculated and evaluated to determine the data variation captured by the model. The PLS predictive method involved tracking the development of the RMSECV and the root mean square error of calibration (RMSEC) parameters, which the rPLS technique extracted and synthesised in relation to the number of estimated LVs.

To determine outliers, clusters, and recognisable patterns in the line plots of the scores, the sample/score plots were constructed. The scatter plots were examined for data points that showed a recurring pattern, and the correlation between the two continuous variables was determined by analysing the point distribution of the scatter plots. On the plot of the measured versus predicted BPA concentration, the form, direction, strength, and presence of outliers were evaluated. The statistical confidence region was added to the scores on the LV1 versus LV2 plot to help identify these outliers because they were outside of it [55]. The total variance in Hotelling’s T^2^ statistic and in the lack-of-fit statistic were obtained from the plot of Hotelling’s T^2^ versus Q residuals reduced. Any significant outliers in the upper- or lower-right corners of the plot were found using the studentized residuals versus leverage plot [56].

The 120 EEM dataset was split into a 20-member validation subset and a 100-member calibration subset. To assess the effectiveness of the PSL model, a PLS analysis was performed on a batch of 100 BPA standards with BPA concentrations ranging from 3 to 300 µM and 20 validation samples with BPA concentrations ranging from 15 to 300 µM. The calibration dataset was used to find or learn relationships between the traits and the target variable. The validation dataset was used to provide an impartial evaluation of model fit on the training dataset while model hyperparameters were being adjusted [57]. BPA concentrations in the spiked surface water samples were predicted based on two-dimensional (2-D) EEMs derived from the unfolding of three-dimensional (3-D) EEMs using principal component analysis.

The performance parameters of the PLS model (RMSECV, RMSEC, coefficient of correlation for prediction (R^2^_Pred_), cross-validation (R^2^_CV_), calibration (R^2^_Cal_), and the root mean square error of prediction (RMSEP or SEP)) were assessed to evaluate the effectiveness of the regression model. A number of figures of merit, including adjusted R-squared (adj. R^2^), standard error, linear fit slope, and intercept parameters, were all established and assessed for the performance of the model in order to further validate it.

#### 3.8.5. PLS Model Validation

The model was shown to be accurate by predicting BPA concentrations in surface water samples that had been spiked. PLS model performance parameters derived from the PLS analysis were used to evaluate the validity of the model. Included in the parameters were the RMSEP, RMSECV, RMSEC, R^2^_Pred_, R^2^_CV_, and R^2^_Cal_.

#### 3.8.6. Validation of Spiked Surface Water Samples

A measured versus predicted BPA concentration plot and a regular residual versus independent variables plot for BPA were constructed using the EEM data from surface water samples spiked with the BPA standards at concentrations of 15 to 300 µM. Origin V8.6 software was used to construct the plots. The predicted and measured BPA concentration plot was used to generate regression parameters, which were then compiled in a report along with other goodness-of-fit parameters. The accuracy of the model was evaluated by comparing the predicted and measured values using RMSE, MAE, Pearson’s r value, adj. R^2^, residual sum of squares, standard error of the intercept, intercept, and slope [41]. The difference between the measured and estimated values assesses how outliers affect the data, with RMSE giving outliers a higher priority than MAE. The standard *p*-value was set at 0.05, and the confidence and prediction intervals were both set at 95%. The plot of the regular residual versus independent variables provided information on the residual distribution. Degrees of freedom, squared sum of residuals, squared mean, F-value, and probability > F or *p*-value statistical parameters were compiled in an ANOVA table.

An F-value was used to determine the statistical significance of the test [58]. Prob > F is the *p*-value for the entire model test. The *p*-value, i.e., the Prob > F-value, was used to ascertain the relationship between the predicted and measured BPA concentrations and test the hypothesis that none of the independent variables are related to the dependent variables.

#### 3.8.7. Method Sensitivity and Limits of Detection and Quantification

The slope of the calibration graph and the standard error of the intercept of the graph were used to calculate the LOD of the A-TEEM-PLS method using Equation (1) [11]:LOD = (3 × Standard error of intercept + Intercept)/Slope(1)

Based on the slope of the calibration graph and the standard error of the intercept of the graph, the LOQ of the A-TEEM-PLS method was calculated using Equation (2):LOQ = (10 × Standard error of intercept + Intercept)/Slope(2)

The LOD or sensitivity is the lowest concentration of a measure that an analytical method can accurately measure. The LOD was therefore the method’s sensitivity in this investigation [46].

#### 3.8.8. Accuracy and Recovery of the Method

The accuracy of the A-TEEM-PLS method was assessed in a single experimental run, with identical solutions, and by a single analyst. The percent recoveries for the new method (Equation (3)) [59] were used to gauge accuracy:% Recovery = (obtained result)/(expected result) × 100(3)

The recoveries were determined through the acquisition of EEM data from surface water samples spiked with BPA at three different concentrations (60, 180, and 270 µM). The results were assessed for compliance with international method validation guidelines (SANCO/12495/2011), which require the mean recovery to range between 70 and 120% [60].

#### 3.8.9. The Robustness of the Model

Because EEMs and component outliers like Raman and Rayleigh scattering have a significant impact on PARAFAC and PLS predictions [31,33], the 3-D EEM dataset underwent preprocessing to eliminate these interferences as described earlier to strengthen the models. The width, smoothness, and similarity of the loading spectra were used to evaluate the robustness of the PARAFAC model. The model was deemed to be robust if the loading spectra were smooth, comparable, broad, and frequently unimodal.

The magnitudes of the coefficients of determination for the calibration (R^2^_Cal_) and cross-validation (R^2^_CV_) parameters were evaluated to determine the effects of nonlinearity. The magnitudes of the R^2^_Cal_, R^2^_CV_, and R^2^_Pred_ parameters were examined in order to gauge the robustness of the PLS model. Tests for nonlinearity effects and outlier removal were carried out to validate the model. As part of the evaluation of the removal of outliers, it was determined how much the correlation coefficients had changed after the outliers were eliminated. LVs were used in the construction of a robust PLS model in order to find pertinent features and/or eliminate irrelevant variables so as to increase prediction accuracy and reduce model complexity.

## 4. Conclusions

The A-TEEM-PARAFAC-PLS analytical method is effective at monitoring BPA contamination in water because of its quick run time (less than 4 min), high sensitivity at lower micro mole levels for BPA, little to no sample preparation, and ease of use. Using multiway-based algorithms in optical data analysis has several benefits, including handling interferents and useful outlier control, because it is difficult to detect aqueous fluorescent substances using optical spectroscopy. Instrument selectivity is not required when using multiway data [10]. BPA was clearly identified by PARAFAC in the presence of uncalibrated interferences in the trilinear experimental data. The obtained results demonstrated that the data from the spectra, extracted and analysed using the PARAFAC and PLS algorithms, could be usefully exploited in the development of PARAFAC and rather robust regression models that demonstrated an ability to identify BPA in surface water and predict the BPA content in surface water.

Future research can investigate combining the Solo regression and discrimination models (method models) developed in this study and multiblock models (absorbance and EEM/PEM data concatenation) to predict BPA concentrations in real surface water using the multimodel predictor tool. The current study could benefit greatly from this improvement in order to move it toward a more advanced stage of predicting BPA concentrations in real surface water analyses, which should at the very least be aimed at achieving reliable classification as a quality control.

Also, A-TEEM spectroscopic methods will need to be developed in the future in conjunction with other chemometric tools, such as decomposition tools (e.g., multivariate curve resolution, SIMPLISMA (purity), and others) and regression tools (e.g., artificial neural network, designed experiment MLR, locally weighted regression, and others), in order to track and quantify water-quality-related parameters like microplastics, turbidity, chemical oxygen demand, and oxidation reduction potential.

## Figures and Tables

**Figure 1 molecules-28-07048-f001:**
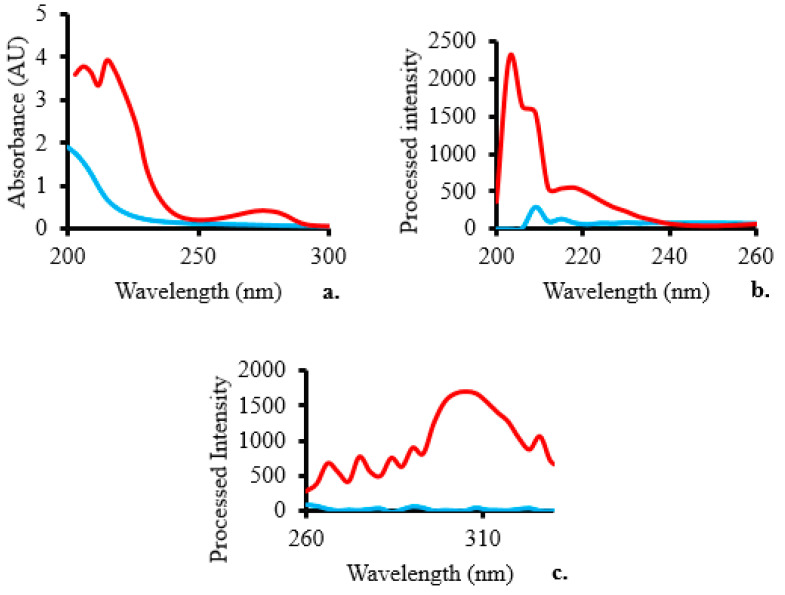
Comparing the baseline (blue line) and a surface water sample spiked with 100 µM BPA (red line) in terms of their respective (**a**) absorbance, (**b**) excitation, and (**c**) emission spectra.

**Figure 2 molecules-28-07048-f002:**
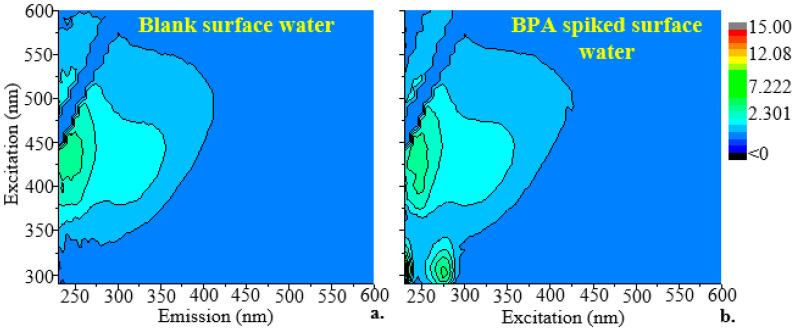
Two-dimensional excitation–emission matrices of (**a**) an unspiked surface water sample with dissolved organic carbon = 2.7 ppm and (**b**) a sample of surface water that had 21 µM of BPA added to it.

**Figure 3 molecules-28-07048-f003:**
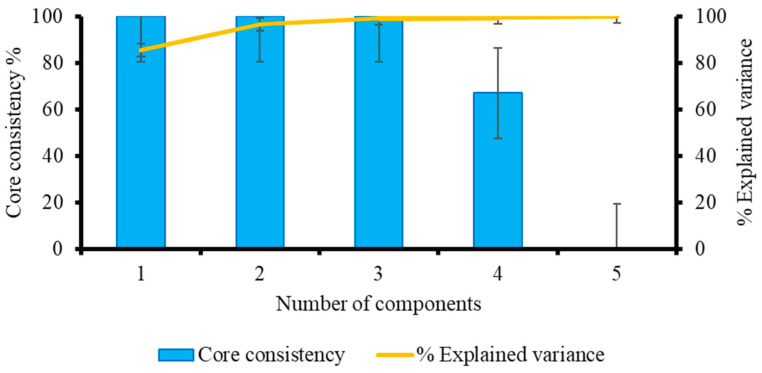
The core consistency diagnostic and explained variance values for the EEM dataset of surface water samples spiked with BPA standard solutions of various concentrations are shown in Figure 3 for the one- to five-component non-negativity-constrained PARAFAC model.

**Figure 4 molecules-28-07048-f004:**
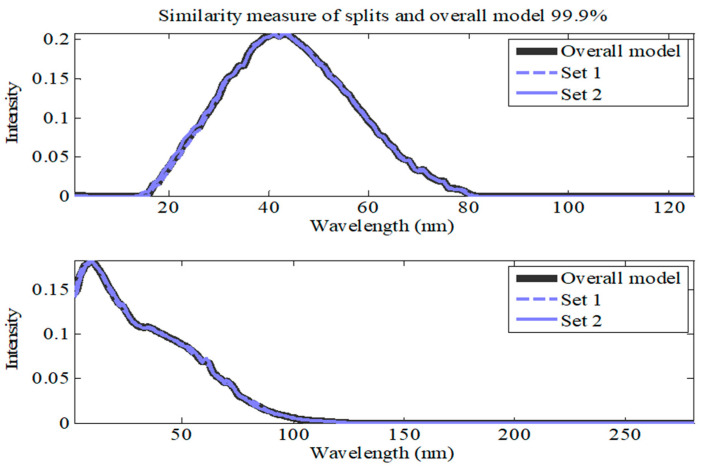
Results of the split-half analysis using the PARAFAC model for EEM data collected from surface water samples spiked with BPA at concentrations ranging from 3 to 300 µM.

**Figure 5 molecules-28-07048-f005:**
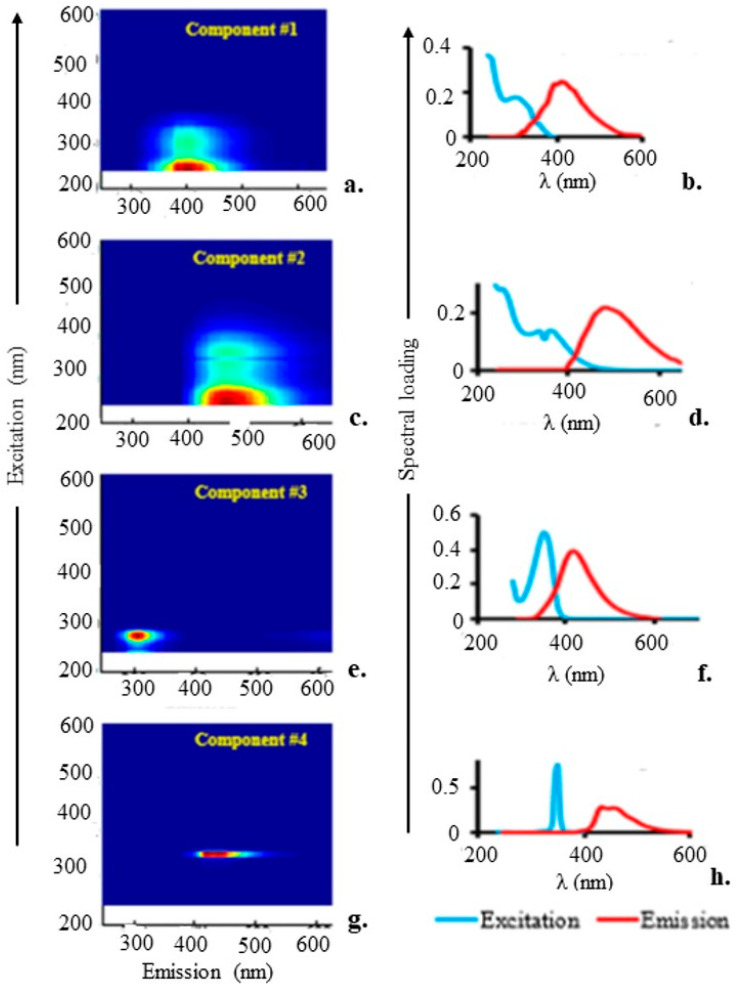
(**a**) Component #1: A 2-D contour plot of the EEM typical of the fulvic acid-like organic matter; (**c**) component #2: A 2-D contour plot of the EEM typical of the humic acid-like fraction of organic matter; (**e**) component #3: A 2-D contour plot of the EEM typical of BPA; and (**g**) component #4: A 2-D contour plot of the EEM typical of the marine humic-like fraction of organic matter. Excitation and emission spectral loadings of the four components obtained via PARAFAC modelling of fluorescence spectral dataset of surface water containing BPA are also shown. (**b**) confirmed the maximum Ex/Em wavelengths for component #1, which correspond to fulvic acid-like organic material and are around 258-323/442 nm, respectively. According to (**d**), the maximum Ex/Em wavelengths for component #2 were around 263-372/504 nm, representing an organic material that resembles humic acid. The maximum Ex/Em wavelengths for component #3, which stands in for BPA, are depicted in (**f**) as being around 274/307 nm. While component #4’s maximum Ex/Em wavelengths are depicted in (**h**) to be approximately 348/435 nm, this component represents organic material that is comparable to marine humic material.

**Figure 6 molecules-28-07048-f006:**
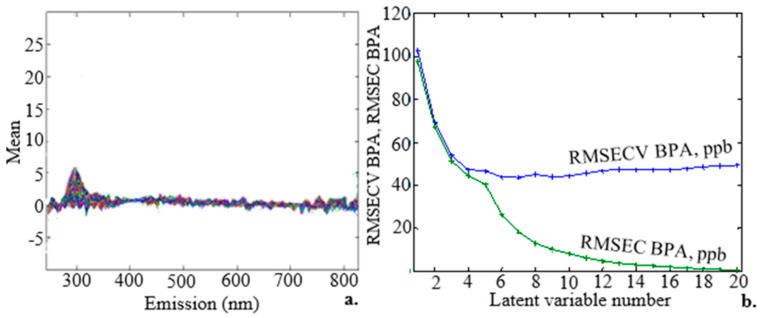
(**a**) Emission spectra for 120 samples of surface water spiked with BPA standards at various but evenly spaced concentrations ranging from 3 to 300 µM. (**b**) The RMSECV (blue curve) and RMSEC (green curve) were created based on the quantity of LVs used to create the prediction model for the 120 EEM spectral data points of BPA. It was possible to obtain at least 5 LVs.

**Figure 7 molecules-28-07048-f007:**
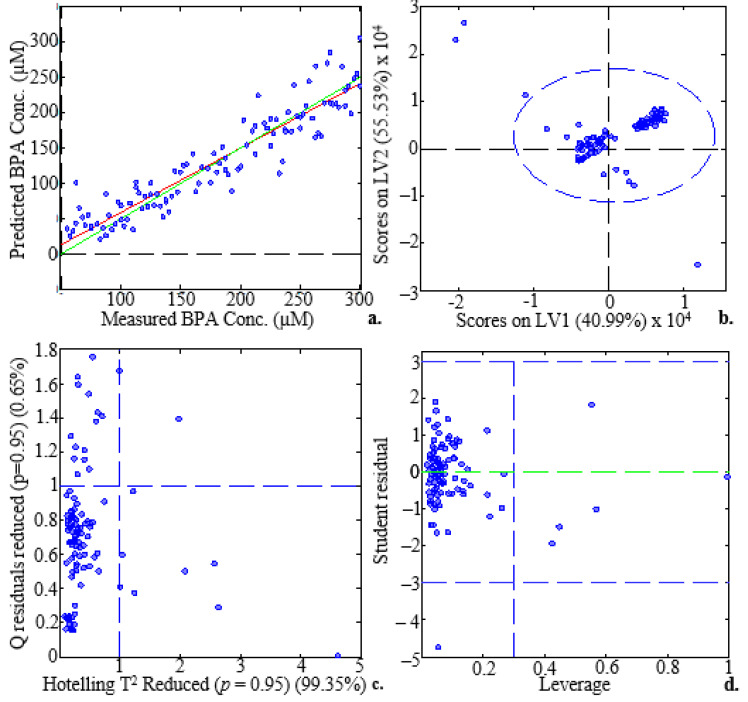
Sample/score plots for the analysis of BPA concentration range (3–300 µM) depicting plots of (**a**) measured BPA (µM) versus predicted BPA concentration (µM). The red line represents a linear fit for the calibration dataset, while the green line represents a linear fit for the validation dataset, (**b**) scores on LV1 versus scores on LV2. The vertical and horizontal lines inside the plot are the x-axis and y-axis zero lines, (**c**) Hotelling’s T^2^ statistic versus Q residuals. The 5% (Q residuals) and 95% (Hotelling’s T2) dashed lines, respectively, show the confidence limits for these variables, and (**d**) leverage versus studentised residuals 1 BPA (µM). The vertical dashed line denotes the warning leverage and the green dashed line denotes the y-axis zero line, while the horizontal upper- and lower dashed lines denote the ±3 studentised residuals.

**Figure 8 molecules-28-07048-f008:**
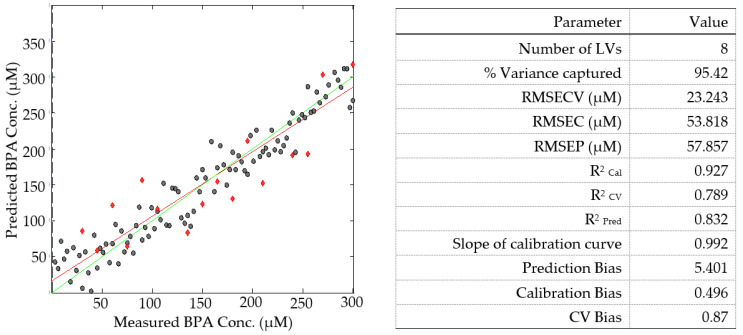
The plot of measured versus predicted BPA concentration (µM) for a 100-member calibration dataset (black dots) and a 16-member validation dataset (red diamonds) and the PLS model performance parameters for the analysis of surface water spiked with BPA. The red line represents the line of best fit with regard to the calibration EEM dataset and the green line represents the line of best fit with regard to the validation EEM dataset.

**Figure 9 molecules-28-07048-f009:**
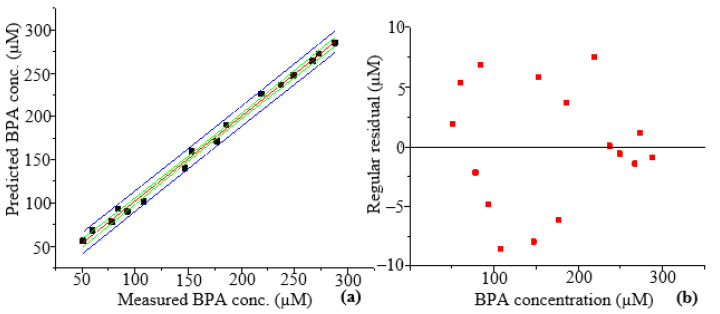
(**a**) The linear fit of the BPA validation data. Green lines show 95% confidence bands, blue lines show 95% prediction bands, and the red line shows the linear fit. The black squares represent the predicted BPA concentration (**b**) The plot of the regular residuals versus the independent variables for BPA. The red squares represent the regular residuals.

**Figure 10 molecules-28-07048-f010:**
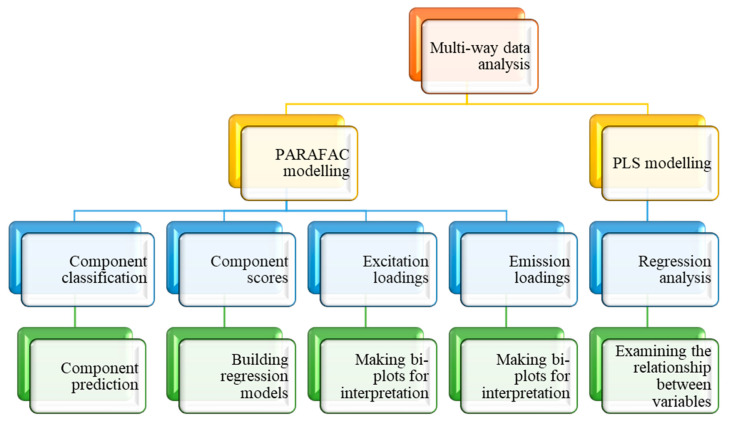
Workflow for multiway EEM data analyses with Solo software (V 9.0).

**Table 1 molecules-28-07048-t001:** Bisphenol A chemical information.

Chemical structure	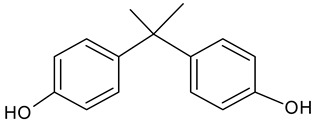
Molecular formula, molecular weight	C_15_H_16_O_2_, 228.291
CAS number	80-05-7

**Table 2 molecules-28-07048-t002:** PLS model performance statistics.

Parameter	Value
Number of LVs	5
RMSEC (µM)	17.434
RMSECV (µM)	34.794
Calibration Bias	1.396
CV Bias	0.33
R^2^ for Calibration	0.967
R^2^ for Cross-Validation	0.845

**Table 3 molecules-28-07048-t003:** The linear regression analysis report for surface water samples spiked with BPA at concentrations of 60, 90, 120, 210, and 300 µM (from Origin V8.6).

Parameter	Value
Residual sum of squares	97.311
Pearson’s r	0.998
R-Squared (COD)	0.996
Adj. R-squared	0.996
RMSE	5.272
MAE	4.378
Intercept	4.219
Standard error of intercept	3.079
Slope	0.98
Standard error of slope	0.0167

**Table 4 molecules-28-07048-t004:** ANOVA table for linearity of the BPA regression model.

	Degrees of Freedom	Sum of Squares	Mean Squares	F Value	Prob > F
Model	1	95,914.648	95,914.648	210.474	0
Error	14	389.073	27.791		
Total	15	96,303.722			

**Table 5 molecules-28-07048-t005:** Percent recoveries of BPA spiked at three different concentration levels in surface water.

Nominal Conc. of BPA (µM)	Measured Conc. of BPA (µM)	Percent Recovery
50	47.715	95.43
180	178.686	99.27
270	264.465	97.95

## Data Availability

Following a valid request, the corresponding author will provide the dataset used and analysed during this study.

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
