# Peer review of "Detection and Quantification of Bisphenol A in Surface Water Using Absorbance–Transmittance and Fluorescence Excitation–Emission Matrices (A-TEEM) Coupled with Multiway Techniques"

_molecules, 2023, doi:10.3390/molecules28207048_

Round 1

Reviewer 1 Report

Dear Authors,

you can find my comments in pdf file attached.

Author Response

Comment

The topic of this article is important and the material and methods part is very well detailed. For me it is questionable, that how good this method is comparing to other analytical methods for BPA determination. I understand, that the method in this article provide a very fast determination and it is a very useful phenomenon, but it would be necessary to highlight the sensitivity of the method.

Response

We changed the subheading of Section 2.6 to: “Method sensitivity and limits of detection and quantification” and wrote that: “Since the lowest concentration of a measure that can be accurately measured by an analytical method is referred to as the lower limit of detection or sensitivity, then the method sensitivity is 3.512 µM. However, the method used in this study had a lower sensitivity than the method used in a prior study to determine BPA at low concentrations in naturally occurring mineral water kept in plastic bottles using HPLC with UV detection [42]. We changed the subheading of Section 3.8.7 to 3.8.7 Method sensitivity and limits of detection and quantification and wrote that: The LOD, or sensitivity, is the lowest concentration of a measure that an analytical method can accurately measure [42]. Therefore, the LOD was the method's sensitivity in this investigation.”

Comment

You apply μM as concentration when you mention the spike level of surface water sample with BPA. However, on some graphs we can read ppb. Please, uniform the concentration units for better comparison? What is the limit of detection of your methods for BPA determination in surface water?

Response

We have changed the units of concentration from ppb to µM on the graphs in Figures 7 and 9.

We wrote that: The calculated detection and quantification limits for BPA in surface water were 3.512 and 11.708 µM, respectively.

Comment

line 52-54: BPA pollution of surface water is a quite big problem. The 273 μM is an average value? If yes, you should indicate standard deviation. This part is the manuscript (BPA concentration in surface water) should be more detailed. You should discuss more articles reporting BPA concentration in surface water worldwide. Please, involve more references here.

Response

We have indicated the standard deviation on 273 μM and added more references and wrote that: it was found that the measured concentration of BPA in surface water ranged from 100 to 400 µM in Europe and North America [13].  In Asia, an upper limit of 0.014 g/L for the concentration of BPA in surface water was recorded [14]. In South African rivers, the concentration of BPA ranged from non-detectable to 89 ng/L [15]. Based on a review of the literature and the distribution of species sensitivities, it was found that the measured concentration of BPA in surface water was 273±9.73 ppb [16]. The median concentration in various surface water samples from Latin American nations was 0.53 ng/mL, while the min-max concentration ranged from 0.09 to 1.46 ng/mL [17].

Comment

line 91: method.at lower > method at lower; micro mole > μM

Response

We changed micro mole to µM.

Comment

line 95: micro mole > μM

Response

We changed micro mole to µM.

Comment

Figure 1: What in the unit of measurement of processed intensity for excitation and emission spectra? What was the detection wavelength for excitation spectra, and what was the wavelength of light source for emission spectra?

Response

Amounts of intensity are expressed in arbitrary units. Therefore, it is acceptable not to specify the intensity units. The excitation spectra of the surface water sample spiked with BPA revealed two peaks at excitation wavelengths of ~203 and ~218 nm. The emission spectra of the surface water sample spiked with BPA revealed a major peak at emission wavelength of ~306 nm.

Comment

line 115: Why did you use 21 μM BPA for spike instead of 100 μM that you applied in obtaining spectra (chapter 2.1)

Response

After evaluating the EEMs of 100 surface water samples for clarity, I decided to settle on the one spiked with 21 µM of BPA because it was very clear at a lower concentration and I wanted to demonstrate the high sensitivity of the method at this low concentration. The one spiked with 100 µM BPA was much darker.

Comment

line 116-117: How are these nm values in connection with the spectra in Figure 1?

Response

The excitation and emission wavelength ranges depicted in Figures 1(b) and (c) are verified in Figure 2 (i.e., lines 116–117), proving that they are identical despite having been obtained through different techniques. Figure 1's (a) describes the relationship between absorbance and wavelength; the wavelengths described in lines 116–117 have no bearing on this plot. BPA's excitation wavelength is shown in (b), which describes the relationship between processed excitation intensity and wavelength, and BPA's emission wavelength is shown in (c), which describes the relationship between processed emission intensity and wavelength. Excitation and emission wavelengths are plotted on the same graph and expressed together in lines 116–117.

Comment

line 125: (b) is duplicated; you should indicate what DOC means.

Response

We deleted the first (b) and remained with one and we wrote DOC in full as: “dissolved organic carbon” 

Comment

Figure 3: What type of data are presented? Mean (blue column) and standard deviation (black lines)?

Response

The core consistency diagnostic and explained variance values for the EEM dataset of surface water samples spiked with BPA standard solutions of various concentrations are shown in Figure 3 for the one- to five-component non-negativity-constrained PARAFAC model.

Comment

Figure 4: What is the unit of measurement for intensity? What is the difference between the two (upper and lower) figures? How many samples did you have between 3 and 300 μM BPA? Which part of the graphs indicates the different BPA concentrations?

Response

The intensity is measured in arbitrary units, like absorbance. So if one does not indicate the units on intensity, it is acceptable.

Comment

line 148-152: Please, write a more detailed description of overall model, set 1 and set 2. In line you mention similarity of 98.8%, however on Figure 4 this number is 99.9%.

Response

We replaced 98.8% by 99.9% and wrote: “The results of the split-half analysis for a four-component model showed a similarity of 99.9% matching for both distinct splits, indicating that almost no difference could be made between the constituents of each separate split set”.

Comment

Figure 5: The resolution of a-b-c-d-e-f-g-h pictures/graphs are very low. What is the unit of measurement of spectral loading?

Response

We enlarged the resolution of a-b-c-d-e-f-g-h pictures and graphs.

A spectral loading has arbitrary units.

Comment

Figure 6: Which part of the graphs indicates the different BPA concentrations? RMSEC is green curve not black.

Response

The different BPA concentrations are hidden in the graphs and are not shown on the graphs. The different BPA concentrations give rise to what is shown by the graphs, i.e., the mean (intensity) and emission spectra on Graph (a), the RMSEC, RMSECV, and the number of latent variables. Graph (a) shows the relationship between the mean (intensity) and emission wavelengths. Graph (b) shows the relationship between the RMSEC, RMSECV, and the number of latent variables.  

Comment

line 224: R2 > R2 (please correct in the whole text)

Response

We replaced R2 with R2

Comment

Figure 8: What are red and green lines on the plot?

Response

The red line represents the line of best-fit with regard to the calibration EEM dataset and the green line represents the line of best-fit with regard to the validation EEM dataset. Thank you, we have added this sentence to caption on Figure 8.

Comment

Figure 9: Please, indicate (a) and (b) on Figure 9. The text in black boxes is unnecessary, because the information is indicated in the description of Figure.

Response

We have indicated (a) and (b) on Figure 9 and removed the black boxes, as shown:

Figure 9: (a) The linear fit of the BPA validation data. Green lines show 95% confidence bands, blue lines show 95% prediction bands, and a red line shows the linear fit. (b) The plot of the regular residuals versus the independent variables for BPA   

Comment

line 337: Text should be bolded.

Response

We have bolded the text in line 337 as “3. Materials and methods” 

Reviewer 2 Report

The manuscript studies an easier method for detection and quantification of bisphenol A (BPA) in surface water using absorbance-transmittance and fluorescence excitation-emission matrices (A-TEEM) coupled with multi-way techniques, which shows some encouraged results for the developing optical techniques for micro-organic pollutants in waters. However, it needs to be carefully revised before it could be acceptable. The main comments are as following:

1.       Actually, this manuscript is not really novel one for measurement of BPA for environmental samples. There are at least three papers related to the study, which are as following:

1)       Excitation-emission fluorescence-kinetic third-order/four-way data: Determination of bisphenol A and nonylphenol in food-contact plastics. Talanta 2019,197:348–355.

2)       Migration test of Bisphenol A from polycarbonate cups using excitation emission fluorescence data with parallel factor analysis. Talanta, 2017,167:367–378.

3)       Simultaneous and fast determination of bisphenol A and diphenyl carbonate in polycarbonate plastics by using excitation-emission matrix fluorescence couples with second-order calibration method. Spectrochimica Acta Part A: Molecular and Biomolecular Spectroscopy, 2019, 216:283–289.

Therefore, the manuscript should not only refer the papers, but also present some advantages or explanation in comparison with these papers.

2.       In Figure 2, it sounds the labels of vertical/ horizontal coordinates have some mistakes. In Figure 4, what is for upper and down figures?

3.       The study should presents some interferes, e.g. PAHs or PCBs coexisting in water samples, to demonstrate the method being reliable.

4.       All text needs to be modified to make it more readable and smooth, from introduction to the result explanations.

There are some language errors, mainly related to grammar.

Author Response

Comment

The manuscript studies an easier method for detection and quantification of bisphenol A (BPA) in surface water using absorbance-transmittance and fluorescence excitation-emission matrices (A-TEEM) coupled with multi-way techniques, which shows some encouraged results for the developing optical techniques for micro-organic pollutants in waters. However, it needs to be carefully revised before it could be acceptable. The main comments are as following:

Response

The manuscript was revised by replacing some text with the following, for example, and we stated in the introduction that "there were three previous studies that used FEEM in conjunction with a number of multi-way techniques to detect and quantify BPA in different plastic materials [4, 5, 6]. In the first prior study, BPA and 4-nonylphenol in samples of different plastic materials that were in contact with beverages and/or food were identified using fluorescence EEM in conjunction with kinetic third-order or four-way PARAFAC data modelling. Four-way PARAFAC modelling provided satisfactory outcomes for BPA analysis. Third-order methods have the advantage of being able to quantify the analytes in the presence of uncalibrated compounds and solve systems with very poor selectivity [7, 8]. The current study combined EEM data from surface water laced with standard solutions of BPA with three PARAFAC analysis modes that were restricted to non-negative values in order to identify BPA and other fluorescent components in surface water laced with BPA. In order to calculate BPA, we used Origin 8.6 to regress the data imported from PLS modelling. The A-TEEM method has the advantage that simultaneous acquisition of absorbance and fluorescence spectral data in a single instrument makes it easier to correct fluorescence IFEs and produce fluorescence spectral data free from IFEs [9].”

We also replaced some text in the introduction with the following in order to make the text readable “While the second prior study [5] proposed a fluorimetric method in conjunction with the second order calibration of EEMs for the evaluation of the BPA migration test, the current study employed the PARAFAC and PLS models, respectively, to detect and quantify BPA in surface water. Due to the trilinearity of the data tensor, which ensured that the solution obtained through PARAFAC analysis was unique, the method used in the previous study was able to identify and quantify BPA with absolute certainty. Consequently, even though other fluorophores were present in the test sample, one element of the decomposition matched up with BPA. The advantage of the PARAFAC model in the current study, however, is that instrument selectivity is not required when using multi-way data [10]. While the current study used the predicted and measured BPA concentration, scores on LV1 versus scores on LV2, and leverage versus studentized residuals plots in addition to the Q Residual versus Hotelling T2 plot for the identification of outlier samples during PARAFAC calibration, this earlier study used the Q Residual versus Hotelling T2 plot only. This suggests that the current study found more outliers overall and a greater variety of them than the earlier study.”

“In comparison to the earlier study [5], PARAFAC models were validated in the current study more thoroughly. This is due to the fact that the previous study used the core consistency diagnostic only to establish the ideal number of components in the developed model, whereas in the current study, the split-half analysis, explained variance, and visualisation of spectral loadings were evaluated in addition to the core consistency diagnostic technique to validate the PARAFAC models. In contrast to this earlier study which quantified BPA using the PARAFAC calibration, the current study used the PLS algorithm to ascertain the concentration of BPA in surface water samples.”

“The third earlier study [6] decomposition algorithm to ascertain the presence of BPA in polycarbonate plastics. Second-order calibration has the benefit of allowing analytes to be quantitated even when there are unmodelled interferents. Instrument selectivity is not required when using multi-way data, which is a benefit of using PARAFAC modelling [10]. As a result, different algorithms were used by the methods to interpret EEM data.”

Please note that the replaced text is more than what has been mentioned in this part of responses.

Comment

Actually, this manuscript is not really novel one for measurement of BPA for environmental samples. There are at least three papers related to the study, which are as following:

1)       Excitation-emission fluorescence-kinetic third-order/four-way data: Determination of bisphenol A and nonylphenol in food-contact plastics. Talanta 2019,197:348–355.

2)       Migration test of Bisphenol A from polycarbonate cups using excitation emission fluorescence data with parallel factor analysis. Talanta, 2017,167:367–378.

3)       Simultaneous and fast determination of bisphenol A and diphenyl carbonate in polycarbonate plastics by using excitation-emission matrix fluorescence couples with second-order calibration method. Spectrochimica Acta Part A: Molecular and Biomolecular Spectroscopy, 2019, 216:283–289.

Therefore, the manuscript should not only refer the papers, but also present some advantages or explanation in comparison with these papers.

Response

We stated in the introduction that "there were three previous studies that used FEEM in conjunction with a number of multi-way techniques to detect and quantify BPA in different plastic materials [4, 5, 6]. In the first prior study, BPA and 4-nonylphenol in samples of different plastic materials that were in contact with beverages and/or food were identified using fluorescence EEM in conjunction with kinetic third-order or four-way PARAFAC data modelling. Four-way PARAFAC modelling provided satisfactory outcomes for BPA analysis. Third-order methods have the advantage of being able to quantify the analytes in the presence of uncalibrated compounds and solve systems with very poor selectivity [7, 8]. The objective of the current study was to characterize BPA and other fluorescent components in surface water laced with BPA by combining EEM data from surface water laced with BPA standard solutions with three PARAFAC analysis modes that were restricted to non-negative values. In order to calculate BPA, we used Origin 8.6 to regress the data imported from PLS modelling. The A-TEEM method has the advantage that simultaneous acquisition of absorbance and fluorescence spectral data in a single instrument makes it easier to correct fluorescence IFEs and produce fluorescence spectral data free from IFEs [9].”

“While the second prior study [5] proposed a fluorimetric method in conjunction with the second order calibration of EEMs for the evaluation of the BPA migration test, the current study employed the PARAFAC and PLS models, respectively, to detect and quantify BPA in surface water. Due to the trilinearity of the data tensor, which ensured that the solution obtained through PARAFAC analysis was unique, the method used in the previous study was able to identify and quantify BPA with absolute certainty. Consequently, even though other fluorophores were present in the test sample, one element of the decomposition matched up with BPA. The advantage of the PARAFAC model in the current study, however, is that instrument selectivity is not required when using multi-way data [10]. While the current study used the predicted and measured BPA concentration, scores on LV1 versus scores on LV2, and leverage versus studentized residuals plots in addition to the Q Residual versus Hotelling T2 plot for the identification of outlier samples during PARAFAC calibration, the earlier study used the Q Residual versus Hotelling T2 plot only. This suggests that the current study found more outliers overall and a greater variety of them than the earlier study.”

“In comparison to the earlier study by [5], PARAFAC models were validated more thoroughly in the current study. This is due to the fact that the previous study used the core consistency diagnostic only to establish the ideal number of components in the developed model, whereas in the current study, the split-half analysis, explained variance, and visualisation of spectral loadings were evaluated in addition to the core consistency diagnostic technique to validate the PARAFAC models. In contrast to this earlier study [5], which quantified BPA using the PARAFAC calibration, the current study used the PLS algorithm to ascertain the concentration of BPA in surface water samples.”

The third earlier study by [6], combined FEEM and second-order chemometrics methods based on the alternating trilinear decomposition algorithm to ascertain the presence of BPA in polycarbonate plastics. Second-order calibration has the benefit of allowing analytes to be quantitated even when there are unmodelled interferents. Instrument selectivity is not required when using multi-way data, which is a benefit of using PARAFAC modelling [10]. As a result, different algorithms were used by the methods to interpret EEM data.

Comment

In Figure 2, it sounds the labels of vertical/ horizontal coordinates have some mistakes. In Figure 4, what is for upper and down figures?

Response

The Aqualog® spectrometer assigned the labels to the vertical and horizontal coordinates in Figure 2 during the analysis. The instrument displayed the labels in this manner.

By dividing the EEM dataset (in relation to samples) into two equal, separate datasets, the split-half validation analysis is performed, and the two halves are independently modelled using PARAFAC. This analysis measures the consistency of the scores of a test, and the results of the test are compared in terms of similarity. The results of the split-half validation of each half of the EEM data based on the number of components that fit into the PARAFAC model are shown in Figure 4. The upper and lower figures are comparisons of the emission spectra from each half of the EMM dataset. The similarity score of the splits is shown at the top of the Figure after analysis is finished.

Comment

The study should present some interferes, e.g. PAHs or PCBs coexisting in water samples, to demonstrate the method being reliable.

Response

We wrote in the method Section that “A green analytical chemistry protocol was applied to solve a problem involving a weakly fluorescent system with significant component emission spectrum overlap. Because methanol has the potential to change the solution's refractive index and contains a fluorescent background, it is called for, among other things, to limit the final methanol concentration in the samples to less than 2% in the cuvette [11].”

We had written in the method section line 446 that “To rectify any systematically biased data, get rid of interference from IFEs, Raman scatter, and Rayleigh scatter, and normalise datasets with substantial intensity variations between samples, the raw EEM data were preprocessed.”

We have written in the conclusion that “When there are uncalibrated interferences present in experimental data that are trilinear, BPA can be clearly identified by PARAFAC.”

In the conclusion we wrote that “Using multi-way-based algorithms in optical data analysis has several benefits, including handling interferents and useful outlier control, because it is difficult to detect aqueous fluorescent substances using optical spectroscopy. Actually, instrument selectivity is not required when using multi-way data [10].”

Comment

All text needs to be modified to make it more readable and smooth, from introduction to the result explanations.

Response

We stated in the introduction that "there were three previous studies that used FEEM in conjunction with a number of multi-way techniques to detect and quantify BPA in different plastic materials [4, 5, 6]. In the first prior study, BPA and 4-nonylphenol in samples of different plastic materials that were in contact with beverages and/or food were identified using fluorescence EEM in conjunction with kinetic third-order or four-way PARAFAC data modelling. Four-way PARAFAC modelling provided satisfactory outcomes for BPA analysis. Third-order methods have the advantage of being able to quantify the analytes in the presence of uncalibrated compounds and solve systems with very poor selectivity [7, 8]. The current study combined EEM data from surface water laced with standard solutions of BPA with three PARAFAC analysis modes that were restricted to non-negative values in order to identify BPA and other fluorescent components in surface water laced with BPA. In order to calculate BPA, we used Origin 8.6 to regress the data imported from PLS modelling. The A-TEEM method has the advantage that simultaneous acquisition of absorbance and fluorescence spectral data in a single instrument makes it easier to correct fluorescence IFEs and produce fluorescence spectral data free from IFEs [9].”

We also replaced some text in the introduction with the following in order to make the text readable “While the second prior study [5] proposed a fluorimetric method in conjunction with the second order calibration of EEMs for the evaluation of the BPA migration test, the current study employed the PARAFAC and PLS models, respectively, to detect and quantify BPA in surface water. Due to the trilinearity of the data tensor, which ensured that the solution obtained through PARAFAC analysis was unique, the method used in the previous study was able to identify and quantify BPA with absolute certainty. Consequently, even though other fluorophores were present in the test sample, one element of the decomposition matched up with BPA. The advantage of the PARAFAC model in the current study, however, is that instrument selectivity is not required when using multi-way data [10]. While the current study used the predicted and measured BPA concentration, scores on LV1 versus scores on LV2, and leverage versus studentized residuals plots in addition to the Q Residual versus Hotelling T2 plot for the identification of outlier samples during PARAFAC calibration, the second earlier study used the Q Residual versus Hotelling T2 plot only. This suggests that the current study found more outliers overall and a greater variety of them than the earlier study.”

“In comparison to the earlier study by [5], PARAFAC models were validated more thoroughly in the current study. This is due to the fact that the previous study used the core consistency diagnostic only to establish the ideal number of components in the developed model, whereas in the current study, the split-half analysis, explained variance, and visualisation of spectral loadings were evaluated in addition to the core consistency diagnostic technique to validate the PARAFAC models. In contrast to the study [5], which quantified BPA using the PARAFAC calibration, the current study used the PLS algorithm to ascertain the concentration of BPA in surface water samples.”

“The third earlier study [6], combined FEEM and second-order chemometrics methods based on the alternating trilinear decomposition algorithm to ascertain the presence of BPA in polycarbonate plastics. Second-order calibration has the benefit of allowing analytes to be quantitated even when there are unmodelled interferents. Instrument selectivity is not required when using multi-way data, which is a benefit of using PARAFAC modelling [10]. As a result, different algorithms were used by the methods to interpret EEM data.”

Please note that the replaced text is more than what has been mentioned in this part of responses.
